# Application of a 72 h National Early Warning Score and Incorporation with Sequential Organ Failure Assessment for Predicting Sepsis Outcomes and Risk Stratification in an Intensive Care Unit: A Derivation and Validation Cohort Study

**DOI:** 10.3390/jpm11090910

**Published:** 2021-09-13

**Authors:** Chih-Yi Hsu, Yi-Hsuan Tsai, Chiung-Yu Lin, Ya-Chun Chang, Hung-Cheng Chen, Yu-Ping Chang, Yu-Mu Chen, Kuo-Tung Huang, Yi-Hsi Wang, Chin-Chou Wang, Meng-Chih Lin, Wen-Feng Fang

**Affiliations:** 1Division of Pulmonary and Critical Care Medicine, Department of Internal Medicine, Kaohsiung Chang Gung Memorial Hospital, Chang Gung University College of Medicine, Kaohsiung 83301, Taiwan; hsuchiyi@gmail.com (C.-Y.H.); flyninesun@gmail.com (Y.-H.T.); chiungyu@cgmh.org.tw (C.-Y.L.); yachun1026@gmail.com (Y.-C.C.); chc1106@cgmh.org.tw (H.-C.C.); b9002087@cgmh.org.tw (Y.-P.C.); blackie@cgmh.org.tw (Y.-M.C.); jelly@cgmh.org.tw (K.-T.H.); yihsi@cgmh.org.tw (Y.-H.W.); ccwang52@cgmh.org.tw (C.-C.W.); mengchih@cgmh.org.tw (M.-C.L.); 2Department of Anesthesiology, Kaohsiung Chang Gung Memorial Hospital, Chang Gung University College of Medicine, Kaohsiung 83301, Taiwan; 3Department of Respiratory Therapy, Kaohsiung Chang Gung Memorial Hospital, Chang Gung University College of Medicine, Kaohsiung 83301, Taiwan; 4Department of Respiratory Care, Chang Gung University of Science and Technology, Chiayi 61363, Taiwan

**Keywords:** severe sepsis, intensive care unit, mortality risk stratification, National Early Warning Score 2, Sequential Organ Failure Assessment

## Abstract

We investigated the best timing for using the National Early Warning Score 2 (NEWS2) for predicting sepsis outcomes and whether combining the NEWS2 and the Sequential Organ Failure Assessment (SOFA) was applicable for mortality risk stratification in intensive care unit (ICU) patients with severe sepsis. All adult patients who met the Third International Consensus Definitions for Sepsis and Septic Shock criteria between August 2013 and January 2017 with complete clinical parameters and laboratory data were enrolled as a derivation cohort. The primary outcomes were the 7-, 14-, 21-, and 28-day mortalities. Furthermore, another group of patients under the same setting between January 2020 and March 2020 were also enrolled as a validation cohort. In the derivation cohort, we included 699 consecutive adult patients. The 72 h NEWS2 had good discrimination for predicting 7-, 14-, 21-, and 28-day mortalities (AUC: 0.780, 0.724, 0.700, and 0.667, respectively) and was not inferior to the SOFA (AUC: 0.740, 0.680, 0.684, and 0.677, respectively). With the new combined NESO tool, the hazard ratio was 1.854 (1.203–2.950) for the intermediate-risk group and 6.810 (3.927–11.811) for the high-risk group relative to the low-risk group. This finding was confirmed in the validation cohort using a separated survival curve for 28-day mortality. The 72 h NEWS2 alone was non-inferior to the admission SOFA or day 3 SOFA for predicting sepsis outcomes. The NESO tool was found to be useful for 7-, 14-, 21-, and 28-day mortality risk stratification in patients with severe sepsis.

## 1. Introduction

Sepsis caused by a dysregulated host response to infection could lead to life-threatening organ dysfunction. Mortality ranges from 29.2 to 33.2% in the USA, UK, and Taiwan [1,2,3]. Sepsis is highly prevalent in the ICU: 18.0% of patients in the ICU were already diagnosed with sepsis at ICU admission [4].

According to the 2016 Third International Consensus Definitions for Sepsis and Septic Shock (Sepsis-3) guidelines, the Sequential Organ Failure Assessment (SOFA) was included in the definition of sepsis and became a part of practice in the intensive care unit (ICU) setting. To date, the SOFA has been widely used to evaluate the severity of sepsis. However, the SOFA requires detailed laboratory data. To save time, the quick SOFA (qSOFA) was then developed for screening patients with suspected infection. Patients with a qSOFA score ≥ 2 points are considered at risk of poor outcomes outside the ICU. Nevertheless, the qSOFA was inferior to SOFA for outcome prediction [5]. For better predictive performance for mortality, the change of SOFA score (ΔSOFA) between different days was assessed and ΔSOFA on day 7 is a useful early prognostic marker of 28-day mortality [6].

In addition to these two indices, other early warning scores were proposed. The National Early Warning Score 2 (NEWS2), which was approved by the National Health Service in the UK, is an updated version of the NEWS, which was reported in December 2017. The NEWS is able to detect acute illness and clinical deterioration in patients early by using their accessible vital signs and clinical status. A study in 2015 indicated that the NEWS can be the trigger to detect and screen patients at risk for septic shock [7].

Recently, many studies have demonstrated that the NEWS2 provided better discrimination of outcomes than the qSOFA, particularly in the emergency department and prehospital setting [8,9,10,11,12]. However, few studies have verified the prognostic accuracy of the NEWS2 for patients that are admitted to the ICU with severe sepsis [13]. Nannan Panday and associates investigated the efficacy of NEWS2 in an acute medical unit and good value was shown [14]. Additionally, the best timing for applying the NEWS2 and the possibility of using it to predict in-hospital mortality has not been fully clarified.

Thus, this study investigated the best timing for applying the NEWS2 for predicting in-hospital mortality. Moreover, we proposed a new tool, which is the combination of the 72 h NEWS2 and the admission SOFA (i.e., NESO) to facilitate in-hospital mortality risk stratification in patients with sepsis within 28 days after ICU admission. We hypothesized that patients with a high SOFA score at ICU admission and a high NEWS2 at the 72 h follow-up had a poor outcome that was related to severe sepsis and a poor response to treatment.

## 2. Materials and Methods

### 2.1. Setting

The study was approved by the institutional review board of Chang Gung Memorial Hospital and the requirement for obtaining patient consent was waived (IRB no. 202001696B0C502). This study was conducted at the Kaohsiung Chang Gung Memorial Hospital, which is a 2700-bed tertiary teaching hospital in Southern Taiwan. The patients were recruited from 3 medical ICUs (total of 34 beds). Data from this study population were also analyzed for other purposes in previous studies [15,16,17].

### 2.2. Study Design

This was a retrospective study that consisted of all consecutive ICU patients who initially qualified as a derivation cohort by meeting the criteria of Sepsis-3 between August 2013 and January 2017. Patients who died within 3 days of ICU admission and those without complete clinical data were excluded (Figure 1a). Finally, 699 patients were analyzed. The primary outcomes were the 7-day, 14-day, 21-day, and 28-day mortalities. The validation cohort consisted of patients with the same setting as above from January 2020 to March 2020.

### 2.3. Data Collection

For both cohorts, we collected the laboratory and clinical data necessary for calculating the SOFA and NEWS2 scores on days 1 and 3 from medical records. Clinical data included age, sex, scoring indices, Charlson Comorbidity Index, underlying comorbidities, and other potentially related clinical and laboratory data. We also collected the clinical data necessary for determining the NEWS2 at exactly 24, 72, and 168 h after ICU admission. The vital signs on day 3 were all reviewed to calculate the highest NEWS2 on that day as the worst day 3 NEWS2.

### 2.4. Scoring Indices

The Acute Physiological Assessment and Chronic Health Evaluation II (APACHE II) model has been used to classify illness severity and predict hospital mortality since 1985 [18]. The qSOFA score was proposed for use in patients that are highly suspected of sepsis and septic shock, where 2 of the following 3 criteria need to be fulfilled: respiratory rate ≥ 22/min, altered mentation, or systolic blood pressure ≤ 100 mmHg [19]. The SOFA score, reflecting grading for 6 vital systems (respiratory, coagulation, cardiovascular/circulatory, liver, central nervous, and renal systems), can identify sepsis among patients with suspected infection. The NEWS2 was calculated using 6 simple physiological parameters, including respiration rate, oxygen saturation, blood pressure, heart rate, consciousness, and body temperature. Additionally, the NEWS2 was modified via integration with the SOFA as a modified NEWS2 by replacing the oxygen saturation of the NEWS2 with the respiratory scale used in the SOFA because, although the same scale was used, the SOFA version precisely reflects the oxygenation capacity.

Given the individual mortality prediction ability of the initial SOFA and 72 h NEWS2, we used a combination of the SOFA and NEWS2 (the new NESO tool) to classify patients into 3 groups: a low-risk group (admission day (Adm) SOFA < 11 and 72 h NEWS2 < 7), an intermediate-risk group (Adm SOFA ≥ 11 but 72 h NEWS2 < 7, or Adm SOFA < 11 but 72 h NEWS2 ≥ 7), and a high-risk group (Adm SOFA ≥ 11 and 72 h NEWS2 ≥ 7).

### 2.5. Statistical Analysis

The values are presented as a frequency and percentage for categorical variables or as mean ± standard deviation for continuous variables with a normal distribution. We used the Statistical Package for the Social Sciences software version 22.0 (IBM Corp., Armonk, NY, USA) for most statistical analyses and generating plots. Group comparisons were conducted using the Pearson chi-squared test, independent *t*-test, and one-way analysis of variance, as appropriate. For survival analysis, Kaplan‒Meier curves and a Cox proportional hazard model were used. The performance of the NEWS2 and SOFA scores was assessed by comparing the receiver operating characteristic (ROC) curve for mortality via the DeLong method with MedCalc version 19.1.3 (MedCalc Software Ltd., Ostend, Belgium). Youden’s index was used to determine the best cutoff values. Statistical significance was set at a two-sided *p*-value of < 0.05.

## 3. Results

### 3.1. Enrolled Background

Overall, 3377 patients were admitted to three ICUs between August 2013 and January 2017. We excluded patients without a diagnosis of sepsis (*n* = 2578), those who died within 3 days after admission to the ICU (*n* = 49), and those with incomplete clinical data needed for the calculation of the NEWS2 and SOFA score at admission and on day 3. Finally, 699 patients with sepsis were enrolled as a derivation cohort (Figure 1a) and 473 patients were enrolled as a validation cohort in our study (Figure 1b).

### 3.2. Verification and Comparison of Scoring Indices at Admission and on Day 3 for Predicting Mortality

The scoring indices that are shown in Appendix A were calculated from the laboratory and clinical data extracted at the closest time point to ICU admission or the first record in the ICU. Admission SOFA (Adm SOFA) performed well and consistently for predicting the 7-day, 14-day, 21-day, and 28-day mortalities (SOFA >10, AUC: 0.629; SOFA > 11, AUC: 0.625; SOFA > 11, AUC: 0.600; and SOFA > 12, AUC: 0.593), although it was only superior to the admission qSOFA (Adm qSOFA). The day 3 SOFA or day 3 NEWS2 were assessed based on the patient’s vital signs around 8:00 a.m. and blood tests on day 3 in the ICU. For both the NEWS2 and SOFA, discrimination was generally greater on day 3 than that at admission (Appendix A). Furthermore, as shown in Table 1 and Table 2, the modified day 3 NEWS2 improved the AUC of the day 3 NEWS2 (AUC: 0.668 vs. AUC: 0.649, *p* = 0.03) for 28-day mortality.

### 3.3. NEWS2 at Different Time Points for Mortality Prediction

The potential of the NEWS2 for outcome prediction was reproduced. More detailed time points for the NEWS2 assessment are summarized in Table 1, Table 2 and Table 3 demonstrates that the NEWS2 at 72 h performed well for the mortality prediction, although it was not significantly different from the AUC of the day 3 NEWS2 or that of the modified day 3 NEWS2, except for a comparison of 14-day mortality (pairwise comparison of ROC curve for mortality within 14 days: 0.006 and 0.018, respectively). Discrimination of the 72 h NEWS2 for the 28-day mortality was not inferior to any NEWS2 assessment in Table 1. For the day 3 NEWS2, the modified day 3 NEWS2, and the 72 h NEWS2, a cutoff of 6 points predicted 14-, 21-, and 28-day mortalities.

Because of the fluctuation of vital signs that typically occurs in the ICU, the worst physiological parameters were selected for the worst day 3 NEWS2. Consequently, the scores for all patients were elevated, as were the cutoff points. A consistent cut-off point of 11 could still be observed, though a comparison of the ROC curves to the 72 h NEWS2 (*p* = 0.660, 0.435, 0.660, and 0.804 for 7-, 14-, 21, and 28-day mortalities, respectively) and the modified day 3 NEWS2 (*p* = 0.293, 0.108, 0.303, and 0.096 for 7-, 14-, 21, and 28-day mortalities, respectively) showed no significant difference (Table 2 and Table 3).

### 3.4. Contribution of the NESO Tool for Risk Stratification: Clinical Features, Laboratory Data, Severity, and Mortality

Based on the findings above, the Adm SOFA and 72 h NEWS2 both seemed to be reliable for mortality prediction. Patients were then tentatively classified into three different risk groups: low-, intermediate-, and high-risk. Table 4 shows the demographic and clinical information for the three risk groups after admission to the ICU. Sex, body mass index (BMI), number of comorbidities, Charlson Comorbidity Index, comorbidities other than coronary artery disease, and use of noninvasive positive-pressure ventilators and intermittent positive-pressure ventilation revealed no significant difference between the three groups. Surprisingly, age groups and percentage of various comorbidities were not significantly different between the three risk groups. The only difference observed was noted for the percentage of DNR: the high-risk group had a higher proportion of patients with a DNR order prior to ICU admission.

The assessments of different score systems and laboratory data are also presented in Table 4. The NEWS2, APACHE II, Adm SOFA, and day 3 SOFA scores differed significantly between the three stratified groups. The APACHE II scores for the three groups were as follows: low risk, 22.86 ± 7.83; moderate risk, 24.99 ± 8.38; and high risk, 27.47 ± 7.76. However, no significant difference was observed in the Adm white blood cell (WBC) count, day 3 WBC count, and Adm segmented neutrophil-to-monocyte (SeMo) ratio. Nevertheless, an elevation in the SeMo ratio was observed only in the high-risk group and was higher on day 3 than that in the low- or intermediate-risk groups. The low-risk group had the lowest C-reactive protein level on days 1 and 3. Thus, risk stratification could be reflected by the severity of the inflammation index.

As shown in Appendix A and Figure 2a, the mortality rates differed between the three groups at 7, 14, 21, and 28 days. In the high-risk group, 49 of 87 patients (44.9%) died within 28 days. Figure 2b demonstrates three separate survival lines. In Table 5, the crude hazard ratio of the intermediate-risk group compared to the low-risk group was 2.334 (adjusted ratio: 1.884). The crude hazard ratio was 6.810 for the high-risk group (adjusted ratio: 5.361).

The influence of various covariates on survival were analyzed using Cox regression analysis. The covariates were selected because of their influence on risk stratification in the previous analyses, as depicted in the abovementioned tables. The DNR status, the variation of SeMo ratio, the day 3 WBC counts, our NESO tool, and the 168 h NEWS2 were all independent predictive factors for survival.

### 3.5. Validation of the NESO Tool and Potential for Other Predictions

The NESO tool for risk stratification was then verified using a validation cohort, which consisted of 473 patients who were filtered using the same inclusion criteria as the derivation group between January 2020 and March 2020 (Figure 1b). In Figure 3, three separate 28-day survival lines could still be observed through the risk stratification and a significant difference existed between the three groups at 7, 14, 21, and 28 days. Figure 3a and Appendix A demonstrate that 28-day mortality for low-, intermediate-, and high-risk groups were 11.0, 28.4, and 54.7%, respectively. For the high-risk group, 28-day mortality was high in either the derivation group or validation group (56.3 and 54.7%, respectively). From the validation cohort, the patients in the low-risk group had fewer days in the ICU and a shorter time of relying on mechanical ventilation, as shown in Appendix A.

## 4. Discussion

The NEWS2 was originally designed for identifying critically ill patients. Given its advantages of utilizing accessible clinical parameters, previous studies proposed that its application should be extended to mortality prediction [8,9,20]. In the present study, the NEWS2 was applied for the assessment of patients that were admitted to the ICU due to severe sepsis. We calculated the time-course of the NEWS2 and evaluated the ability of these scores to predict mortality. We found that the early NEWS2 did not reflect the outcomes of patients, but the NEWS2 calculated after 72 h was able to discriminate outcomes (Table 1). By combining the Adm SOFA and the 72 h NEWS2, we stratified patients into different risk groups, and a significant separation of the survival curves between the three groups was found. The NESO tool (Adm SOFA + 72 h NEWS2) had a greater AUC for the ROC curve (AUROC) than the Adm SOFA or APACHE II alone for 28-day mortality prediction from the validation cohort (0.704 vs. 0.656, *p* = 0.03; 0.704 vs. 0.633, *p* = 0.02). This stratification also reflected the severity of inflammation and immunoparalysis. Moreover, the NESO tool could estimate the 7, 14, 21, and 28-day mortality risks for ICU patients with severe sepsis.

To date, many researchers have attempted to verify the ability of different scoring systems to predict the mortality of patients with sepsis or suspected sepsis [9,21,22]. The qSOFA or systemic inflammatory response syndrome (SIRS) criteria have typically been used due to their convenience and reliable sensitivity. The SIRS, qSOFA, and NEWS were all standardized track-and-trigger systems. The NEWS was first proposed in 2012 by the Royal College of Physicians, which was even earlier than the proposal of the qSOFA. The NEWS differed from the above systems most notably in that it was used in people with acute illness rather than in individuals with suspected infection. Accordingly, most studies on the NEWS recruited patients from the ED [9,20,21]. Nevertheless, the NEWS showed equivalent or superior discrimination to the SIRS and qSOFA for in-hospital mortality. A meta-analysis by Fernando et al. in 2018 showed the poor accuracy of the qSOFA, with a sensitivity of 60.8% for predicting mortality within 30 days in adult patients with suspected infection, although it was not compared with the NEWS [23]. The potential of the NEWS warrants discussion.

For ICU patients with infection or suspected infection, a 2017 study revealed that the SOFA had a greater ability for discriminating in-hospital mortality than the SIRS or qSOFA [24]. Nevertheless, in any context, the prognostic accuracy of the SOFA was not doubted. In most previous studies, evaluation scores were based on data that was collected during emergency department visits or within 24 h after ICU admission. A cohort study demonstrated that the NEWS2 that was evaluated from triage for any cause in the emergency department could predict early mortality within 48 h [25]. In our study, the ability of the SOFA, NEWS2, and modified NEWS2 for predicting mortality within 7, 14, 21, and 28 days was analyzed at different time points. As expected, the Adm SOFA had a greater AUC for the ROC curve (AUROC) than the Adm NEWS2 at all time points. The Adm NEWS2 was not able to predict 7-day mortality. Therefore, the original NEWS2 was slightly modified by improving the precision of the respiratory parameters. However, no improvement was revealed when comparing the Adm NEWS2 and modified Adm NEWS2.

Nevertheless, the AUROC values in our study were all markedly lower than those previous studies. Khwannimit et al. reported that the AUROC of the NEWS2 score for mortality within 30 days, which was calculated using the worst value of physiological data within 24 h after ICU admission, was 0.876 [13]. A study performed in 2009 reported that the SOFA score evaluated at 72 h after ICU admission demonstrated greater accuracy for mortality prediction than the SOFA score calculated in the emergency department [26]. We obtained similar results in our study. In addition, the NEWS2 also benefited from a delayed evaluation; the day 3 NEWS2 or 72 h NEWS2 performed better than the Adm SOFA and was not inferior to the day 3 SOFA in terms of mortality prediction. Compared with the SOFA, the NEWS2 was easier to track. Therefore, re-evaluating patients again after 72 h via the NEWS2 was feasible.

We propose a combination of the Adm SOFA and 72 h NEWS2 for risk stratification. Although the original concept for the SOFA was to evaluate the extent of organ function, rather than for mortality prediction, Ferreira et al. showed that stratification using the SOFA at ICU admission correlated roughly with mortality [27]. In their data, the mortality rate for patients with a SOFA score > 11 exceeded 90%. In addition, other studies showed that a SOFA score at admission > 11 could predict mortality in more than 80% of cases [28,29]. The cutoff SOFA score for 14- and 21-day mortalities was also 11 in our study (Appendix A).

On the other hand, we also verified the consensus by the Royal College of Physicians for the NEWS2, which concluded that a score of 7 or greater was a key trigger for activating critical care competencies. However, we set the cutoff point for the admission day SOFA score at 10, based on the lowest Youden Index from the 7-, 14-, 21-, and 28-day mortality predictions. The Adm SOFA and 72 h NEWS2 both showed good performances regarding mortality prediction. Therefore, a combination of the SOFA and NEWS2 could be reliable and easy to implement.

Among sepsis patients, age was shown to be an independent predictor of mortality [25]. However, the age groups were not significantly different between the three risk stratification groups. In our study, there was also no significant influence of sex, BMI, number of comorbidities, and Charlson Comorbidity Index on mortality risk, although the Charlson Comorbidity Index correlated with in-hospital mortality [30]. In terms of morbidity, the proportion of the various comorbidities in the three groups were similar. Nevertheless, the patients in the higher-risk group had a greater proportion of DNR statuses. This might reflect the more critical initial condition, which could have led to a worse outcome.

In Table 4, our stratifications corresponded to increased APACHE II scores. A higher APACHE II score was related to more severe disease and a higher risk of death [31]. Although we stratified our patients using a combination of the Adm SOFA and 72 h NEWS2, the distribution of the Adm NEWS2 was significantly different between the risk groups. The result indirectly showed that the NEWS2 could recognize the risk of mortality and deterioration at the initial ICU admission in patients with severe sepsis.

The SeMo ratio reflects the response of the host immune system. Our previous study showed that the SeMo ratio was related to 28-day mortality in sepsis patients [17]. The dynamic SeMo ratio could also predict mortality. Fang et al. concluded that patients with delta SeMo (day 3 SeMo—Adm SeMo) ≥ 7 had poor clinical outcomes [10]. In the present study, we verified this, where the mean delta SeMo in the high-risk group was 7.49 ± 34.55. CRP is a prognostic marker: nonsurvivors have a higher CRP concentration than survivors [32]. Our study showed that patients in the low-risk group had the lowest CRP level. Nevertheless, the CRP level on day 3 was not an independent predictor of mortality based on Cox regression analysis (Table 5). Although Wang et al. found that WBC count was an independent predictor of 28-day mortality in patients with infection in the emergency department, there was no significant difference between our three groups, regardless of whether this was assessed on the admission day or day 3 [33]. Our NESO tool showed significantly separated survival lines, and the above results for prognostic markers verified this, even though our stratification tool did not include these markers.

In our opinion, a high Adm SOFA score and a high 72 h NEWS2 afterward indicated severe sepsis and poor response to current treatment. Because fluctuating clinical conditions are usually found in ICU patients, it is hard to predict a patient’s outcome by arbitrarily using a single initial clinical assessment. In the other study from our research, combining another clinical assessment (namely, the change in AKI status) as dynamic monitoring could be used as a sepsis phenotype to predict hospital mortality [34]. We verified the power of our tool for mortality prediction for different days. The low-risk group had significantly higher survival rates. Statistically significant differences were revealed in pairwise comparisons for 7-, 14-, 21-, and 28-day mortalities. The original stratification using the NEWS2 (high NEWS ≥ 7) by the Royal College of Physicians was also verified in our study. Although the value of applying biomarkers for personalized prediction was emphasized recently, the effect and cost of these are still being discussed [35].

The study had certain strengths. First, the patients enrolled in this study were consecutive and were all admitted to the ICU due to severe sepsis. Therefore, detailed vital signs and laboratory data were collected, which allowed for a complete analysis. The NESO tool is a novel method that could easily be applied in the ICU based on continuous monitoring without the need for follow-up laboratory data. We expect that our NESO tool could help to identify patients in greater danger, which could then facilitate appropriate treatment or closer monitoring of their clinical condition. No extra effort was needed to apply this tool for ICU staff in clinical practice. For patients in this study, the average time from ICU admission to discharge was 30.4 days. Making a risk assessment within 72 h can still help clinicians to adjust ongoing medication and explain the deterioration to a patient’s family. The sensitivity and specificity for predicting 28-day mortality showed that high-risk stratification in the NESO tool had 38% sensitivity and 91.9% specificity. That is, patients who were not in the high-risk group had a very high chance for a better outcome within 4 weeks. Most importantly, sepsis is a potentially life-threatening condition. Though the sensitivity was not high, this indicated that the clinicians should always treat all patients with sepsis in ICU carefully, especially patients that are considered at high risk by NESO.

However, the study also had some limitations. First, this study was retrospective. Although patients were cared for in a large, high-quality medical center, selection bias is still possible in a single-center study. The results might differ in a smaller local hospital setting. Nevertheless, the tool could be useful in such hospitals lacking medical and human resources. This tool (NESO) is currently an experimental design and was verified only for predicting the prognosis of sepsis within 28 days. Instead of using it for mortality predictions, it will be interesting to further find out whether more active treatment (e.g., real-time hemodynamic monitoring or antimicrobial therapy escalation) is beneficial for high-risk patients. We hope to discover its clinical usage in a future prospective study as a decision tree. In the future, prospective studies with a validation cohort would be needed to verify this tool.

## 5. Conclusions

Though the SOFA score provided a good prediction for sepsis-related mortality, the 72 h NEWS2 was not inferior to the Adm SOFA or day 3 SOFA for predicting the outcome of sepsis patients that were admitted to the ICU. The NEWS2 assessment is easy; therefore, it could also be used for a quick follow-up to identify patients with high mortality risk. The NESO tool can be applied for 7-, 14-, 21-, and 28-day mortality risk stratification in patients with severe sepsis. We propose the use of the NESO tool for early prediction of the outcome of severe sepsis in an ICU.

## Figures and Tables

**Figure 1 jpm-11-00910-f001:**
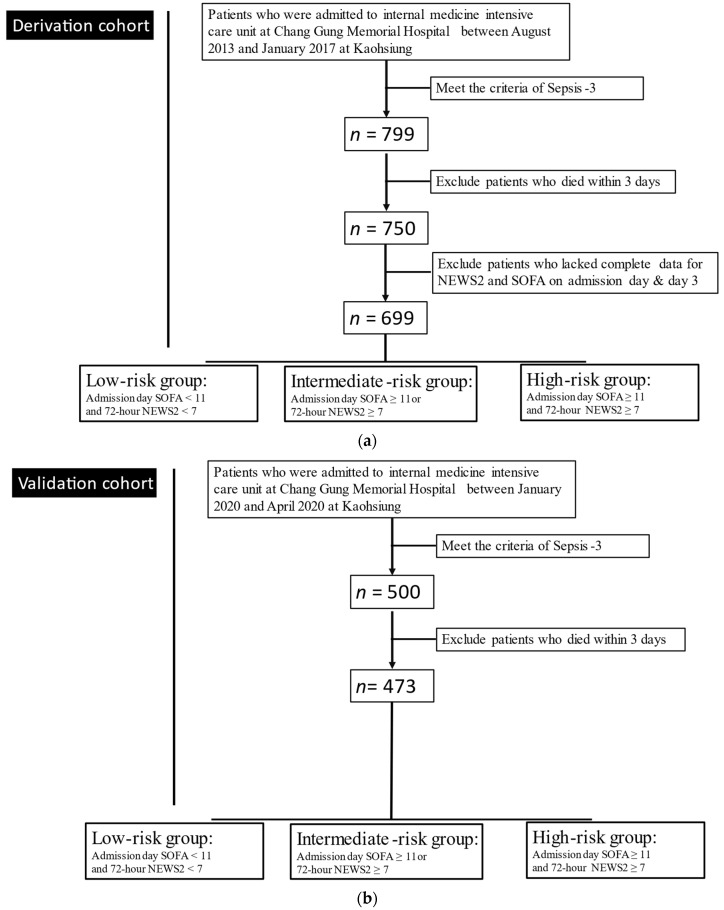
Study flowchart: (**a**) derivation cohort; (**b**) validation cohort.

**Figure 2 jpm-11-00910-f002:**
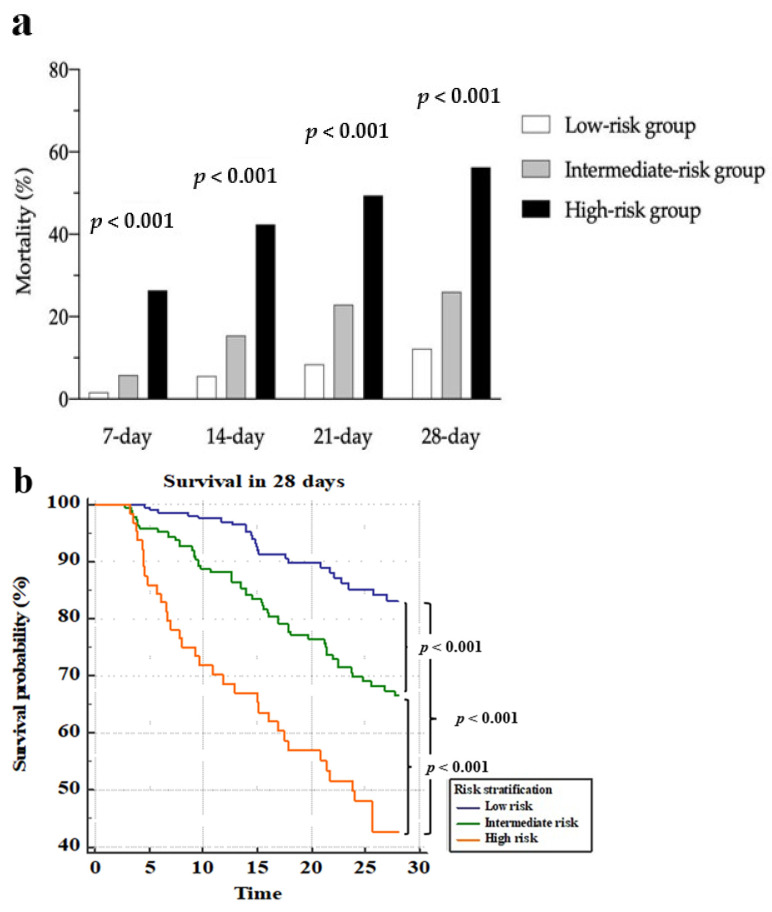
Mortality and survival curves over 28 days in the derivation cohort via the NESO tool. (**a**) Mortality within 7, 14, 21, and 28 days among the low-risk, intermediate-risk, and severe-risk groups, which were stratified using the admission SOFA and the 72 h NEWS2; (**b**) survival curves for the three groups, which were stratified using the admission SOFA and the 72 h NEWS2.

**Figure 3 jpm-11-00910-f003:**
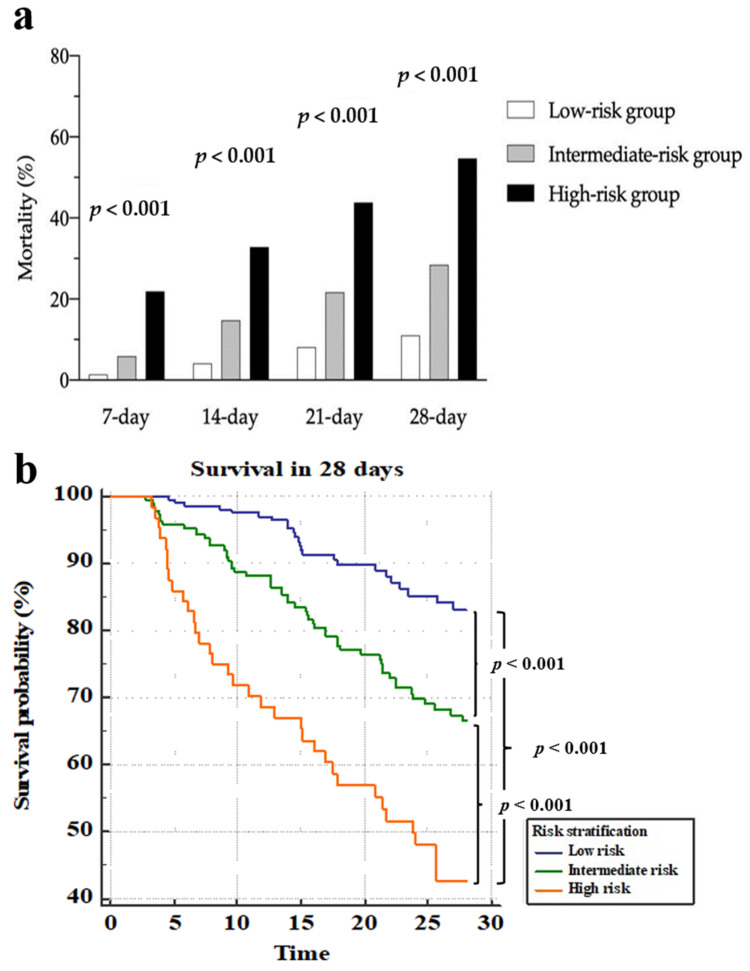
Mortality and survival curves over 28 days in the validation cohort via the NESO tool. (**a**) Mortality within 7, 14, 21, and 28 days among the low-risk, intermediate-risk, and severe-risk groups, which were stratified using the admission SOFA and the 72 h NEWS2; (**b**) survival curves among the three groups, which were stratified using the admission SOFA and the 72 h NEWS2.

**Table 1 jpm-11-00910-t001:** The 7-, 14-, 21-, and 28-day mortality predictions according to 8 prediction rules in all sepsis patients from the derivation cohort.

Prediction Rules	Cut-Off Values	AUC (95%CI)
Admission NEWS2		
7-day mortality	>8	0.566 *
14-day mortality	>8	0.568 *
21-day mortality	>11	0.511 *
28-day mortality	>11	0.569 *
Day 3 SOFA		
7-day mortality	>11	0.740 *
14-day mortality	>10	0.680 *
21-day mortality	>8	0.684 *
28-day mortality	>8	0.677 *
Day 3 NEWS2		
7-day mortality	>8	0.741 *
14-day mortality	>6	0.657 *
21-day mortality	>6	0.669 *
28-day mortality	>6	0.649 *
Modified Day 3 NEWS2		
7-day mortality	>8	0.754 *
14-day mortality	>6	0.664 *
21-day mortality	>6	0.678 *
28-day mortality	>6	0.668 *
24 h NEWS2		
7-day mortality	>6	0.645 *
14-day mortality	>5	0.627 *
21-day mortality	>6	0.631 *
28-day mortality	>6	0.617 *
72 h NEWS2		
7-day mortality	>9	0.780 *
14-day mortality	>6	0.724 *
21-day mortality	>6	0.700 *
28-day mortality	>6	0.667 *
168 h NEWS2		
7-day mortality	>12	0.919 *
14-day mortality	>10	0.749 *
21-day mortality	>9	0.695 *
28-day mortality	>10	0.647 *
Worst Day 3 NEWS2		
7-day mortality	>11	0.718 *
14-day mortality	>11	0.704 *
21-day mortality	>11	0.690 *
28-day mortality	>11	0.672 *

* Significance level. NEWS2: National Early Warning Score 2, SOFA: Sequential Organ Failure Assessment.

**Table 2 jpm-11-00910-t002:** Pairwise comparison of receiver operating characteristic curves for 7-day and 14-day mortalities (the number represents the *p*-value).

	Admission NEWS2	Day 3 SOFA	Day 3 NEWS2	Modified Day 3 NEWS2	24 h NEWS2	72 h NEWS2	168 h NEWS2	Worst Day 3 NEWS2
Admission NEWS2		0.001	0.0004	0.0001	0.032	0.0001	0.0001	0.0007
Day 3 SOFA	0.003		0.985	0.773	0.061	0.1832	0.0002	0.666
Day 3 NEWS2	0.008	0.556		0.275	0.068	0.018	0.0001	0.520
Modified Day 3 NEWS2	0.004	0.671	0.528		0.038	0.406	0.0003	0.293
24 h NEWS2	0.031	0.345	0.346	0.241		0.011	0.0001	0.096
72 h NEWS2	0.0001	0.230	0.006	0.018	0.001		0.002	0.660
168 h NEWS2	0.0001	0.084	0.005	0.011	0.001	0.448		0.0001
Worst Day 3 NEWS2	0.0001	0.075	0.075	0.108	0.007	0.433	0.205	

**Table 3 jpm-11-00910-t003:** Pairwise comparison of receiver operating characteristic curves for 21-day and 28-day mortalities (the number represents the *p*-value).

	Admission NEWS2	Day 3 SOFA	Day 3 NEWS2	Modified Day 3 NEWS2	24 h NEWS2	72 h NEWS2	168 h NEWS2	Worst Day 3 NEWS2
Admission NEWS2		0.002	0.005	0.002	0.034	0.0001	0.537	0.0001
Day 3 SOFA	0.0006		0.678	0.857	0.088	0.622	0.369	0.843
Day 3 NEWS2	0.009	0.375		0.358	0.199	0.146	0.225	0.402
Modified Day 3 NEWS2	0.0009	0.773	0.03		0.111	0.303	0.0003	0.599
24 h NEWS2	0.030	0.042	0.254	0.065		0.017	0.859	0.024
72 h NEWS2	0.0009	0.748	0.367	0.096	0.064		0.861	0.660
168 h NEWS2	0.0251	0.378	0.952	0.493	0.346	0.492		0.880
WorstDay 3 NEWS2	0.0002	0.879	0.299	0.842	0.026	0.804	0.410	

**Table 4 jpm-11-00910-t004:** Demographic characteristics and comorbidities in risk groups from the derivation cohort.

	Total	Low-Risk	Intermediate-Risk	High-Risk	*p*-Value
N	699	320	292	87	
Age	67.37 ± 14.94	68.22 ± 14.93	66.50 ± 15.11	67.21 ± 14.40	>0.05
Sex (Male)	411	57.2%	60.3%	59.8%	0.762
BMI	22.68 ± 4.86	22.67 ± 4.68	22.43 ± 5.26	23.54 ± 3.95	>0.05
DNR	257	84 (26.3%)	119 (40.8%)	54 (62.1%)	<0.001 ^a^
Number of comorbidities	1.70 ± 1.18	1.70 ± 1.18	1.69 ± 1.17	1.74 ± 1.25	>0.05
Charlson Comorbidity Index	2.56 ± 1.96	2.51 ± 1.95	2.57 ± 1.97	2.71 ± 1.96	>0.05
CAD	182 (26.0%)	93 (29.1%)	70 (24.0%)	19 (21.8%)	0.227
Hypertension	395 (56.6%)	191 (59.7%)	160 (54.8%)	44 (51.2%)	0.264
COPD	105 (15%)	58 (18.1%)	35 (12.0%)	13 (13.8)	0.099
Asthma	26 (3.7%)	10 (3.1%)	13 (4.5%)	3 (3.4)	0.515
Pulmonary TB	54 (7.7%)	24 (7.5%)	23 (7.9%)	7 (8.0%)	0.978
Malignancy	158 (22.7%)	69 (21.6%)	67 (23.2%)	22 (25.3%)	0.749
HBV	25 (3.6%)	10 (3.1%)	10 (3.4%)	5 (5.7%)	0.497
HCV	34 (4.9%)	12 (3.8%)	16 (5.5%)	6 (6.9%)	0.675
Cirrhosis	56 (8%)	18 (5.6%)	27 (9.2%)	11 (12.6%)	0.060
DM	315 (45.1%)	146 (45.6%)	134 (45.9%)	35 (40.2%)	0.624
CVA	130 (18.6%)	55 (17.2%)	61 (20.9%)	14 (16.1%)	0.408
CKD	218 (31.2%)	96 (30.0%)	87 (29.8%)	35 (40.2%)	0.129
Intubation	639 (91.4%)	292 (91.3%)	264 (90.4%)	83 (95.4%)	0.341
NIPPV	29 (4.1%)	11 (3.4%)	15 (5.1%)	3 (3.4%)	0.540
APACHE II	-	22.86 ± 7.83	24.99 ± 8.38	27.47 ± 7.76	<0.05
Adm NEWS2	-	6.81 ± 3.05	8.41 ± 2.95	9.82 ± 3.19	<0.001 ^a^
24 h NEWS2	-	5.60 ± 2.22	7.60 ± 2.52	8.53 ± 2.53	<0.001 ^a^
168 h NEWS2	-	7.73 ± 4.29	8.44 ± 3.66	10.13 ± 3.82	<0.05 ^c^
Adm SOFA	-	7.83 ± 3.08	9.03 ± 3.59	12.86 ± 2.83	<0.001 ^a^
Day 3 SOFA	-	6.20 ± 2.17	7.81 ± 2.97	13.94 ± 2.75	<0.001 ^a^
Adm WBC (K)	-	14.24 ± 8.11	14.47 ± 8.03	13.62 ± 8.95	>0.05
Day 3 WBC (K)	-	12.33 ± 6.97	13.14 ± 6.28	13.78 ± 7.45	>0.05
Adm SeMo ratio	-	29.22 ± 35.72	30.30 ± 28.92	31.03 ± 26.09	>0.05
Day 3 SeMo ratio	-	25.05 ± 21.79	27.00 ± 31.18	36.91 ± 30.13	<0.05 ^c^
Day 3 Adm SeMo ratio	-	−4.57 ± 38.82	−3.35 ± 39.34	7.49 ± 34.55	0.027 ^c^
Adm CRP	-	126.49 ± 107.27	155.61 ± 119.11	160.60 ± 126.00	0.012 ^b^
Day 3 CRP	-	106.67 ± 90.79	136.34 ± 100.02	167.28 ± 113.97	<0.05 ^a^

All values are given as mean (SD) or number and percent. BMI, body mass index; DNR, do not resuscitate order; CAD, coronary artery disease; COPD, chronic obstructive pulmonary disease; TB, tuberculosis; HPV, human papillomavirus; HCV, hepatitis C virus; DM, diabetes mellitus; CVA, cerebrovascular accident; CKD, chronic kidney disease; Adm, admission; WBC, white blood cell; SeMo, segmented neutrophil-to-monocyte ratio; CRP, C-reactive protein. ^a^ Significant difference between measurements. ^b^ No significant difference between intermediate-risk group and high-risk group. ^c^ High-risk group > intermediate-risk group = low-risk group.

**Table 5 jpm-11-00910-t005:** Crude and adjusted hazard ratios for mortality in the derivation cohort.

Variable	HR	95%CI of HR	*p*-Value
Crude			
Risk stratification †			
Intermediate-risk group	2.344	1.698–3.236	<0.001
High-risk group	6.810	3.927–11.811	<0.001
Adjusted			
Risk stratification †			
Intermediate-risk group	1.884	1.203–2.950	0.023
High-risk group	5.361	2.704–7.521	0.002
Variation of SeMo ratio (day 3 Adm)	1.004	1.000–1.008	0.027
DNR	3.382	2.300–4.972	<0.001
Day 3 WBC	1.030	1.003–1.057	0.027
Day 3 CRP	0.999	0.997–1.001	0.309
168 h NEWS2	1.145	1.093–1.199	<0.001

† Reference category: low-risk group. BMI, body mass index; DNR, do not resuscitate order; CAD, coronary artery disease; COPD, chronic obstructive pulmonary disease; TB, tuberculosis; HPV, human papillomavirus; HCV, hepatitis C virus; DM, diabetes mellitus; CVA, cerebrovascular accident; CKD, chronic kidney disease; Adm, admission; WBC, white blood cell; SeMo, segmented neutrophil-to-monocyte ratio; CRP, C-reactive protein.

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
