# Peer review of "Application of a 72 h National Early Warning Score and Incorporation with Sequential Organ Failure Assessment for Predicting Sepsis Outcomes and Risk Stratification in an Intensive Care Unit: A Derivation and Validation Cohort Study"

_jpm, 2021, doi:10.3390/jpm11090910_

Round 1
Reviewer 1 Report
The authors evaluated different scores and the combination of these scores for mortality prediction in septic patients. The authors combined a new prediction tool for patients at high risk on day 3 after ICU admission. Is this tool better than SOFA-Score on admission to the ICU? What is the additional benefit compared to SOFA on admission? To improve the outcome of patients with sepsis an efficient therapy must be established within the first hours after ICU admission. How could this tool improve the outcome of sepsis patients if it can only be used on day 3 after admission?
Reviewer 2 Report
The authors used developmental and validation cohorts of patients with sepsis to study the combination of NEWS2 and SOFA score in stratification of the risks of mortality. Although the overall design is contemporary, the investigation has not generated any new tool to clinical usage. The aims and conclusions are very vague.
Even the validation is not convincing clinician to adapt this method. NEWS 72h scores and SOFA score cannot give early warning of the risks for patients. Moreover, combination of two sets of scores will take much more time of medical workers. Therefore, this is not something that immediately benefits clinical management.
There is no comparison with other score systems, not even properly compare the NEWS2 with SOFA scores, so it is very hard to conclude that NEWS2 or NEWS2+SOFA is a better choice.
Minor points
- The authors missed some important relevant reference for NEWS2.
- The Table 1b and table 1c are not necessary
- Table 2 shall be Table 1 and Table 1a shall be Table 2
- Figure 2 and Figure 3 are not standard presentation
- Need statistician to comment on the statistical analysis
Round 2
Reviewer 1 Report
The authors have adequately addressed my comments.
Author Response
Thank you for reading patiently. This article was much improved at your suggestion.
Reviewer 2 Report
The authors have answered most of the question and modified the manuscript accordingly. It would be more clear if the authors could discuss the specificity and sensitivity of NESO.
